# Anti-SARS-CoV-2 Activity of Extracellular Vesicle Inhibitors: Screening, Validation, and Combination with Remdesivir

**DOI:** 10.3390/biomedicines9091230

**Published:** 2021-09-16

**Authors:** Supasek Kongsomros, Ampa Suksatu, Phongthon Kanjanasirirat, Suwimon Manopwisedjaroen, Somsak Prasongtanakij, Kedchin Jearawuttanakul, Suparerk Borwornpinyo, Suradej Hongeng, Arunee Thitithanyanont, Somchai Chutipongtanate

**Affiliations:** 1Department of Pediatrics, Faculty of Medicine Ramathibodi Hospital, Mahidol University, Bangkok 10400, Thailand; supasek.kon6@gmail.com (S.K.); suradej.hon@mahidol.ac.th (S.H.); 2Department of Microbiology, Faculty of Science, Mahidol University, Bangkok 10400, Thailand; ampa.sus@mahidol.ac.th (A.S.); swiboonut@gmail.com (S.M.); 3Excellent Center for Drug Discovery (ECDD), Faculty of Science, Mahidol University, Bangkok 10400, Thailand; phongthon.kan@mahidol.ac.th (P.K.); kedchin.jer@student.mahidol.ac.th (K.J.); bsuparerk@gmail.com (S.B.); 4Office of Research, Academic Affairs and Innovation, Faculty of Medicine Ramathibodi Hospital, Mahidol University, Bangkok 10400, Thailand; somsak.pra@mahidol.ac.th; 5Department of Biotechnology, Faculty of Science, Mahidol University, Bangkok 10400, Thailand; 6Department of Clinical Epidemiology and Biostatistics, Faculty of Medicine Ramathibodi Hospital, Mahidol University, Bangkok 10400, Thailand; 7Chakri Naruebodindra Medical Institute, Faculty of Medicine Ramathibodi Hospital, Mahidol University, Bang Phli 10540, Thailand

**Keywords:** antiviral, calpeptin, combination, COVID-19, extracellular vesicles, inhibitors, SARS-CoV-2, remdesivir

## Abstract

The coronavirus disease 2019 (COVID-19) pandemic severely impacts health, economy, and society worldwide. Antiviral drugs against SARS-CoV-2 are urgently needed to cope with this global crisis. It has been found that the biogenesis and release mechanisms of viruses share a common pathway with extracellular vesicles (EVs). We hypothesized that small molecule inhibitors of EV biogenesis/release could exert an anti-SARS-CoV-2 effect. Here, we screened 17 existing EV inhibitors and found that calpeptin, a cysteine proteinase inhibitor, exhibited the most potent anti-SARS-CoV-2 activity with no apparent cytotoxicity. Calpeptin demonstrated the dose-dependent inhibition against SARS-CoV-2 viral nucleoprotein expression in the infected cells with a half-maximal inhibitory concentration (IC50) of 1.44 µM in Vero-E6 and 26.92 µM in Calu-3 cells, respectively. Moreover, calpeptin inhibited the production of infectious virions with the lower IC50 of 0.6 µM in Vero E6 cells and 10.12 µM in Calu-3 cells. Interestingly, a combination of calpeptin and remdesivir, the FDA-approved antiviral drug against SARS-CoV-2 viral replication, significantly enhanced the anti-SARS-CoV-2 effects compared to monotherapy. This study discovered calpeptin as a promising candidate for anti-SARS-CoV-2 drug development. Further preclinical and clinical studies are warranted to elucidate the therapeutic efficacy of calpeptin and remdesivir combination in COVID-19.

## 1. Introduction

Coronavirus disease 2019 (COVID-19), caused by severe acute respiratory syndrome coronavirus 2 (SARS-CoV-2), has affected 194 million people worldwide, with 4 million deaths as of 27 July 2021 [1]. COVID-19 vaccines have been rolled out to control the ongoing global pandemic [2,3,4,5,6]. However, there have been recent reports among the complete vaccinated individuals with the breakthrough infection [7,8,9]. The emergence of new SARS-CoV-2 variants (i.e., the Delta variant) can lead to evade vaccine-induced neutralizing antibodies [10,11,12], posing a threat to worsen the current COVID-19 situation.

Antiviral therapy is considered as a mainstay treatment for patients with severe COVID-19, regardless of variants. Drug repurposing is the fastest way to find antiviral agents during the outbreak than the de novo drug discovery process [13,14]. The World Health Organization (WHO) expert group recommended that remdesivir, a nucleotide analog inhibitor of viral RNA-dependent RNA polymerase (RdRp) for treating Ebola virus, was considered to be the priority among the repurposed antiviral drugs, based on in vitro and in vivo against SARS-CoV-2 [15,16]. However, remdesivir was found to have marginal efficacy in hospitalized patients with severe COVID-19 [17]. To address this issue, a combinatorial regimen of repurposing drugs that inhibit multiple processes in the viral replication cycle might be an attractive strategy to improve the effectiveness of COVID-19 therapeutics.

Extracellular vesicles (EVs) are non-replicated membrane-enclosed nanoscale vesicles released from all living cells into extracellular space. EV subtypes include small EVs (exosomes; 30–140 nm in diameter) and large EVs (or microvesicles; 200–1000 nm in diameter), which differ in biogenesis and biophysical properties [18,19]. EVs also carry RNA, DNA, proteins, and lipids from the cell of origin [18,19]. EVs have many features resembling the viruses since their crossing path in biogenesis and sharing the cellular vesiculation machinery [20]. The endosomal sorting complexes required for transport machinery (ESCRT) and tetraspanins are involved in both EVs and virions formation [20], and thereby they share striking similarities in lipid composition (high cholesterol and glycosphingolipids) and protein content (e.g., enriched in tetraspanins, ligands and receptors, and cytosolic proteins) [20,21]. RNA viral biogenesis requires the members of the ESCRT complex, such as tumor susceptibility gene 101 (TSG101) and vacuolar protein sorting-associated protein 4 (VPS4), which are the critical elements of EV biogenesis [22,23]. In addition, studies showed that tetraspanins (which are enriched on EV membranes) might participate in coronavirus fusion events [24,25]. The tetraspanin CD9 could form cell-surface complexes with TMPRSS2 to facilitate MERS-CoV entry and infection of mouse lungs in vivo [24]. MERS-CoV and SARS-CoV utilized tetraspanin-enriched microdomains on the host cells to facilitate proteolytic priming and virus–cell membrane fusion [25]. Notably, CD9 proteins are present in small and large EVs and have a critical role in EV biogenesis and cargo sorting [25,26]. It has been proposed that EV biogenesis and release modulations can benefit RNA viral infection, including SARS-CoV-2 [27,28].

Recently, several small molecules have been extensively studied for their roles in blocking the generation and release of EVs, so-called EV inhibitors, in treating various pathological conditions [29]. As virus and EV biogenesis share some common pathways [20], the use of EV inhibitors might be an alternative strategy for treating COVID-19. This study aimed to discover EV inhibitors with anti-SARS-CoV-2 effect. We screened 17 existing EV inhibitors (extensively reviewed by Catalano and O’Driscoll [29]) to evaluate the inhibitory potential against SARS-CoV-2 infection. Calpeptin was recognized as the most potent inhibitor from the primary screening. The anti-SARS-CoV-2 activity of calpeptin was then validated by using the high-content imaging and viral output assay in the standard cell line for studying anti-SARS-CoV-2 agents, Vero E6 [30,31], and the legitimate model of infected human lung epithelial cell, Calu-3 [32]. Finally, a combination of calpeptin and remdesivir demonstrated a synergistic antiviral effect against SARS-CoV-2 infection in vitro.

## 2. Materials and Methods

### 2.1. Cell Culture

Cell lines were obtained from the American Type Culture Collection (ATCC, Manassas, VA, USA). Vero E6 cells (ATCC^®^CRL-1586™), African green monkey (*Cercopithecus aethiops*) kidney epithelial cells, were cultured in DMEM (Gibco, Grand Island, NY, USA) with 10% FBS (Gibco), 100 U/mL penicillin (Gibco), and 100 µg/mL streptomycin (Gibco). Vero cells (ATCC^®^CCL-81™), African green monkey epithelial cells, were cultured in Minimum Essential Medium (MEM) (Gibco) with 10% FBS (Gibco) and 1% L-glutamine (Gibco). Calu-3 cells (ATCC^®^HTB-55™), human lung epithelial cells, were cultured in DMEM/F12 (Gibco) with 100 U/mL penicillin (Gibco), 100 µg/mL streptomycin (Gibco), and supplemented with 10% FBS (Gibco) and 1% GlutaMAX (Gibco). All cells were grown at 37 °C in a 5% CO_2_ atmosphere.

### 2.2. Virus

SARS-CoV-2 (SARS-CoV-2/01/human/Jan2020/Thailand) was isolated in Vero cells from the nasopharyngeal swab of a patient with COVID-19 in Thailand (GenBank: QYZ85362.1) [30,31]. Viral stocks were propagated in Vero E6 cells, as previously described [30,31]. Briefly, the virus was adsorbed onto a monolayer of Vero E6 cells at 37 °C for 1 h before replacing the infection media with 2% FBS (Gibco) in DMEM (Gibco). Infected cells were then incubated at 37 °C, at 5% CO_2_, until a cytopathic effect (CPE) was observed. Virus-containing supernatant was collected as the virus stocks and was titrated in quadruplicate in 96-well microtiter plates on Vero E6 cells in 4-fold serial dilution to obtain 50% tissue culture infectious dose (TCID50). The viral titer was calculated with the Reed and Münch endpoint method. All the experiments with live SARS-CoV-2 virus were conducted at a certified biosafety level 3 facility at the Department of Microbiology, Faculty of Science, Mahidol University, Thailand.

### 2.3. Compounds

d-Pantethine, imipramine, GW4869, calpeptin, Y-27632, imatinib mesylate, sulfisoxazole, bisindolymaleimide I, indomethacin, NSC23766, clopidogrel, glibenclamide, Chloramidine, amiloride, and U0126 were purchased from Selleckchem (Selleckchem, Houston, TX, USA). Manumycin A and cytochalasin D were purchased from Sigma-Aldrich (Sigma-Aldrich, St. Luis, MO, USA). All compounds (purity > 97%) were dissolved in dimethyl sulfoxide (DMSO; Sigma-Aldrich, St. Luis, MO, USA), at a concentration of 50 mM, before use.

### 2.4. In Vitro Anti-SARS-CoV-2 Assays

In vitro anti-SARS-CoV-2 assays were performed as previously described [30,31]. Vero E6 cells were seeded at 1 × 10^4^ cells per well, and Calu-3 cells were seeded at 5 × 10^4^ cells per well in a 96-well black plate (Corning, Corning, NY, USA) and left to adhere overnight at 37 °C, 5% CO_2_. For primary screening, compounds were diluted in a culture medium to achieve a final concentration of 10 µM with DMSO < 0.5%. For dose responses and drug combinations, compounds were diluted in a culture medium to get the desired concentration with DMSO < 0.5%. All the compounds for in vitro drug screenings were prepared before the start of the infection. Cells were washed with phosphate-buffered saline (PBS) and adsorbed with SARS-CoV-2 at 25TCID50 for 2 h at 37 °C. The viral inoculum was then removed, and cells were washed twice with PBS. After cell infection, a fresh culture medium containing the drug at the indicated concentration was added. Positive convalescent serum (heat-inactivated at 56 °C for 30 min) of a COVID-19 patient was used as a positive control for viral inhibition. Cells were then maintained at 37 °C, at 5% CO_2_, for 48 h. After 48 h post-infection, the infected cells were fixed and permeabilized with ice-cold acetone:methanol (1:1) (Sigma-Aldrich) for 20 min and subjected to detect SARS-CoV-2 nucleoprotein (NP) expression, using a high-content imaging system. At the same time, the culture supernatants were collected to quantify the viral output.

#### 2.4.1. High-Content Imaging System for SARS-CoV-2 Nucleoprotein Detection

Following acetone:methanol fixation, the cells were washed with phosphate-buffered saline with 0.5% Tween (PBST) three times. The cells were then blocked by 2% (*w*/*v*) BSA in PBST for 1 h, at room temperature. Cells were incubated with 1:500 rabbit anti-SARS-CoV-2 NP monoclonal antibody (Sino Biological Inc., Beijing, China) at 37 °C for 1 h. Thereafter, cells were washed with PBST three times, followed by incubation with 1:500 of the goat anti-rabbit IgG Alexa Fluor 488 (Thermo Fisher Scientific, Waltham, MA, USA). After washing, cells were stained with Hoechst dye (Thermo Fisher Scientific). The fluorescent signals were detected and analyzed by the high-content imaging system (Operetta, PerkinElmer, Waltham, MA, USA) at 40× magnification. The percentage of the infected cells in each well was automatically obtained from 16 images per well, using Harmony software (PerkinElmer). Data were normalized to the infected control, and the IC50 value was calculated by GraphPad Prism 7 (GraphPad Company, San Diego, CA, USA).

#### 2.4.2. Virus Output Assay

Viral output (or the number of infectious virions released from infected cells) was measured by the plaque assay. The Vero cell monolayer was prepared in a 6-well plate 24 h before infection. The cells were infected with a serial dilution of the virus and incubated for 1 h at 37 °C. Then, the cells were overlaid with 3 mL/well of overlay medium containing MEM supplemented with 5% FBS and 1% agarose. The culture was incubated at 37 °C, in 5% CO_2_, for three days. Plaque phenotypes were visualized by 0.33% Neutral Red staining (Sigma-Aldrich, St. Luis, MO, USA) for 5 h. Plaque numbers were counted as plaque-forming units per milliliter (PFUs/mL) and presented as the percentage of plaque reduction. This experiment was performed in two biological replicates, and the data were presented as the mean of two independent experiments.

### 2.5. Cell Viability Assay

Vero E6 cells were seeded at 1 × 10^4^ cells per well, and Calu-3 cells were seeded at 5 × 10^4^ cells per well in a 96-well plate and allowed to adhere overnight at 37 °C. Cells were treated with serial dilutions of compounds in a medium for 48 h. Cell viability was examined by the MTT colorimetric assay (Sigma-Aldrich). In brief, the medium was replaced with MTT [3-(4,5-dimethylthiazol-2-yl)-2,5-diphenyltetrazolium bromide] at a final concentration of 0.5 mg/mL and incubated for 4 h, at 37 °C, with 5% CO_2_. The MTT solution was removed, and DMSO (Sigma-Aldrich) was added to the cell to dissolve the formazan crystals. Absorbance was measured at a wavelength of 570 nm by an EnVision Multilabel reader (PerkinElmer). Data were normalized to the solvent control, and then 50% cytotoxic concentration (CC50) values were calculated by using GraphPad Prism 7 (GraphPad Company).

### 2.6. Statistical and Data Analysis

The curve was fitted by using non-linear regression, and the IC50 and CC50 values were calculated by GraphPad Prism 7 (GraphPad Company). All statistical tests were performed by using GraphPad Prism version 5. Multiple comparisons were performed by one-way analysis of variance (ANOVA) with the Tukey post hoc test to compare differences among groups.

## 3. Results

### 3.1. Screening of 17 EV Inhibitors for Anti-SARS-CoV-2 Activity

Since EVs and viruses have been found to cross paths in biogenesis [20], it has been postulated that EV inhibitors could serve as the antiviral agents of host-targeting strategy [27,28]. Therefore, this study focused on elucidating whether any of the 17 known EV inhibitors [29] exert an anti-SARS-CoV-2 effect. Chemical structures of EV inhibitors are shown in Figure 1.

The infected Vero E6 cells were used for a primary screen of the 17 EV inhibitors at a fixed concentration of 10 µM. The positive convalescent serum of a COVID-19 patient was included as a positive control for viral inhibitory effects. The criteria were set to identify the hit compound that led to a decrease in viral infection by >50% and had low to modest toxicity (>50% cell viability). As a result, calpeptin was the only EV inhibitor that met the hit compound criteria, achieving up to 95% reduction of the infected cells (Figure 2a,b and Appendix A) with no apparent cytotoxicity at 10 µM (Figure 2c). Notably, cytochalasin D at 10 µM exhibited overwhelming cytotoxicity in Vero E6 cells (Figure 2c and Appendix A). Thus, its anti-SARS-CoV-2 activity should be excluded (Figure 2b). Accordingly, only calpeptin was subjected to further validations of SARS-CoV-2 inhibitory activity.

### 3.2. Calpeptin Exerts the Dose-Dependent Antiviral Activity against SARS-CoV-2 Infection

Next, we validated the hit compound from the primary screening by the dose–response analyses. Vero E6 cells were infected with SARS-CoV-2 at 25TCID50 for 2 h and subsequently treated with calpeptin at the varied concentrations of 0.008 to 100 µM. The anti-SARS-CoV2 activity was determined by high-content imaging of fluorescent NP-positive-infected cells. As a result, calpeptin exerted an anti-SARS-CoV-2 effect in a dose-dependent manner (Figure 3a) with IC50 of 1.44 µM (Figure 3b), while the toxicity of calpeptin, as determined by the MTT assay, showed no apparent cytotoxic with CC50 > 100 µM (Figure 3b). In addition, the viral output study was performed to determine the inhibitory activity of calpeptin against infectious virions released. Calpeptin also exhibited the dose-dependent inhibition of viral output in the culture supernatant of Vero E6 with IC50 of 0.60 µM (Figure 3c). Moreover, the antiviral activity of calpeptin against SARS-CoV-2 was also determined in a human cell line representing the human-lung epithelial cell (Calu-3).

Since human respiratory epithelial lining is the main site of SARS-CoV-2 infection, the dose–response analyses were performed to elucidate whether calpeptin could efficiently inhibit SARS-CoV-2 infection of Calu-3 cells. Accordingly, calpeptin demonstrated the dose-dependent inhibition as determined by high-content imaging of fluorescent NP positive cells with IC50 of 26.92 µM (Figure 4a,b) and quantified by viral output with IC50 of 10.12 µM (Figure 4c). Calpeptin also had no apparent cytotoxicity in Calu-3 cells with CC50 > 100 µM.

### 3.3. Combination of Calpeptin and Remdesivir Increases Antiviral Activity against SARS-CoV-2

The combination of drugs targeting multiple steps in viral life cycle has potential benefits against SARS-CoV-2 infection. In this direction, we further evaluated whether or not calpeptin (possibly targeting viral release) combined with remdesivir (targeting viral replication) exhibited a synergistic anti-SARS-CoV-2 effect. Vero E6 and Calu-3 human lung epithelial cells were infected with SARS-CoV-2 at 25TCID50 for 2 h and subsequently treated with a combination of remdesivir and calpeptin. We decided to evaluate calpeptin (1 and 25 µM for Vero E6 and Calu-3, respectively) and remdesivir (0.5 and 0.1 µM for Vero E6 and Calu-3, respectively) at the concentrations below the IC50 values based on the high-content imaging studies in the corresponding cell lines (Figure 3b and Figure 4b). The dose-dependent effects of remdesivir and IC50 in the infected Vero E6 and Calu-3 cells are provided in Appendix A.

In Vero E6 cells, we found that the combination of calpeptin and remdesivir had a higher anti-SARS-CoV-2 effect than monotherapy, either calpeptin or remdesivir treatment alone, as quantified by %infected cells using the high-content imaging system (Figure 5a,c) and the viral output assay (Figure 5d). Consistently, this synergistic antiviral effect was observed in the infected Calu-3 cells (Figure 5b,f,g). In addition, the combination of calpeptin and remdesivir no apparent cytotoxicity in both Vero E6 and Calu-3 cells (Figure 5e,h). Our results indicated that the combination of calpeptin and remdesivir, even at the suboptimal dosages, substantially enhanced their inhibitory effects against SARS-CoV-2 infection.

## 4. Discussion

EVs have been shown to cross paths in biogenesis with viruses [20]. Therefore, inhibiting EV biogenesis and release may interfere SARS-CoV-2 infection, mainly via the viral budding and production. Our study was performed to explore the anti-SARS-CoV-2 activity of the existing EV inhibitors [29]. Of 17 EV inhibitors screened, calpeptin stands out with the most potent anti-SARS-CoV2 activity, with no apparent cytotoxicity observed (Figure 2). Then, validations of antiviral activity of calpeptin were rigorously performed using two assays (high-content imaging detection of SARS-CoV-2 NP expression and the viral output studies) in two cell lines (Vero E6 kidney cells and Calu-3 lung epithelial cells) in order to confirm SARS-CoV-2 inhibitory activity of calpeptin with high confidence (Figure 3 and Figure 4).

Our findings are in line with previous studies [33,34,35,36,37]. Barnard et al. [33] reported that calpain inhibitors, including calpeptin, could inhibit SARS-CoV replication in vitro. Ma et al. [34] utilized an enzyme kinetic assay to demonstrate that calpeptin could target SARS-CoV-2 M^pro^ viral protease, and thus calpeptin may inhibit viral replication. The IC50 values of calpeptin against SARS-CoV in Vero cells (20 µM) [33] and SARS-CoV-2 M^pro^ (10.69 µM) [34] were shown in a similar trend as the IC50 value reported in this study (1.44 μM), using Vero E6 cells. However, the IC50 value of calpeptin was higher in the infected Calu-3 cells (26.92 μM). This finding might be explained by phenotypic differences between Vero E6 and Calu-3 cells that led to differ antiviral efficacies. Both Vero E6 and Calu-3 cells are susceptible to SARS-CoV-2 infection [30,31], but Vero E6 (monkey kidney) cells lack genes encoding type-I interferons in response to viral infections [38]. In contrast, Calu-3 cells are human-lung epithelial cells with intact immune responses [39]. A recent work by Hoffmann et al. [32] also demonstrated that anti-SARS-CoV-2 activity of small molecules can be cell-type dependent. Chloroquine, which efficiently blocked SARS-CoV-2 viral entry of Vero (kidney) cells, did not appreciably inhibit the SARS-CoV-2 infection of Calu-3 human-lung epithelial cells [32]. Therefore, these results indicated that calpeptin exerted antiviral activity against SARS-CoV-2 in a cell-type independent manner, suggesting that it might be a candidate for developing an effective and safe antiviral agent.

We propose that calpeptin is supposed to mainly inhibit viral release rather than viral replication, as evidenced by the IC50 values of calpeptin based on the viral output studies in both Vero E6 and Calu-3 cells that were >2-fold lower than the IC50 values based on the percentage of infected cells (Figure 3 and Figure 4). Nonetheless, it should be emphasized that this cysteine protease inhibitor may also inhibit coronaviral entry via inhibition of host cysteine protease [35,36]. Recently, Mediouni et al. [37] found that calpeptin may exert dual effects of SARS-CoV-2 inhibition at the viral entry and post-entry processes. Thus, our findings contribute to the current evidence that the post-entry action of calpeptin is responsible by the inhibition of SARS-CoV-2 infectious virion release (Figure 3c and Figure 4c).

To date, little is known about the SARS-CoV-2 egress. SARS-CoV-2 may use the multivesicular body (MVB)-like structure for packaging virions that will be released to the extracellular space via exocytosis [28,40,41,42,43,44]. Alternatively, SARS-CoV-2 may release from infected cells via unconventional egress by lysosomal exocytosis instead of the conventional biosynthetic secretory pathway [42,45], by which it will likely be released as microvesicles [28,43]. SARS-CoV-2 may also be released through a shedding microvesicles bud from infected cells [43,44]. The release of microvesicles can be triggered by Ca^2+^ activating calpain, calcium-dependent cysteine proteases [29]. Microvesicle generation is then inhibited by calpeptin, a protease inhibitor of calpains, as it has been found to reduce bleb formation in hepatocytes [46]. Calpeptin has also been demonstrated to inhibit the shedding of microvesicles in several cell lines [47,48,49]. Moreover, calpeptin has been illustrated to inhibit multivesicular body release [50]. Thus, we propose that calpeptin might inhibit SARS-CoV-2 infection by suppressing viral production through a shared pathway of EV release. Further mechanistic studies are needed to confirm this hypothesis. Nonetheless, the potential viral release inhibition by targeting host-cell machinery of calpeptin is explicitly usable as a good adjunct to the mainstay treatment using the viral replication inhibitors.

In this direction, we then evaluated the combination of calpeptin, the host-targeting inhibitor, and remdesivir, the direct virus-targeting drug authorized for emergency use to treat COVID-19 [13,15,16]. Our results showed that combination treatment of calpeptin with remdesivir boosted the antiviral activity against SARS-CoV-2 compared to monotherapy (Figure 5). This finding supports future investigations of calpeptin and remdesivir combination in preclinical animal models and early phase clinical studies. Our proposed model of this synergistic effect of calpeptin and remdesivir is illustrated in Figure 6.

Since remdesivir has been extensively used for COVID-19 treatment, the virus might develop the resistant strains under the drug pressure, due to the mutations in the RdRp [51]. To address this issue, the combination therapy using drugs targeting both virus and host factors could be an attractive strategy to prevent and overcome resistance. Additionally, targeting host proteins that are shared in a common pathway with viruses can potentially offer a broad-spectrum antiviral effect [52]. However, one should be aware that drugs targeting host cellular proteins have a higher risk of developing toxicities and side effects, as the drugs could interfere with the normal cellular functions [52]. In this regard, we found that the combination of remdesivir and calpeptin, even at the suboptimal dosages, exhibited substantially high inhibitory effects against SARS-CoV-2 infection. This finding allows us to propose the low-dose drug combination strategy, which may have great benefits on the cost-effectiveness and the safety profile, for further preclinical and clinical studies.

This study has several limitations. Calpeptin is an investigational drug and has not been approved for clinical use at this moment. In addition, the study was conducted only in vitro. To address these issues, in vivo animal experiments should be performed to evaluate the efficacy and safety of calpeptin, including the combination with remdesivir. This data will be critical for future research, especially the clinical studies. Lastly, we tested the antiviral activity of calpeptin on SARS-CoV-2 infection and demonstrated the inhibitory effect in Vero E6 and Calu-3 cells; however, in-depth mechanistic studies of calpeptin may be more persuasive and may lead to new therapeutic targets focusing on host proteins.

In conclusion, this study screened the existing EV inhibitors to find calpeptin as a promising candidate for further anti-SARS-CoV-2 development. Combining calpeptin with remdesivir is an attractive treatment strategy that should be pursued in future preclinical and clinical studies.

## Figures and Tables

**Figure 1 biomedicines-09-01230-f001:**
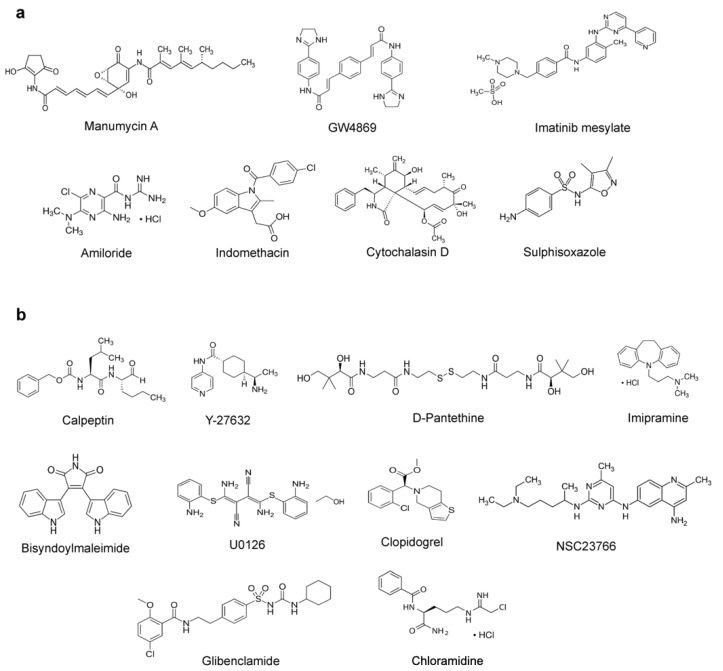
EV inhibitors for screening as anti-SARS-CoV-2 agents in this study. (**a**) Exosome production and release inhibitors. (**b**) Microvesicle generation inhibitors. Targets and mechanism of action (MOA) of these compounds are provided in Appendix A.

**Figure 2 biomedicines-09-01230-f002:**
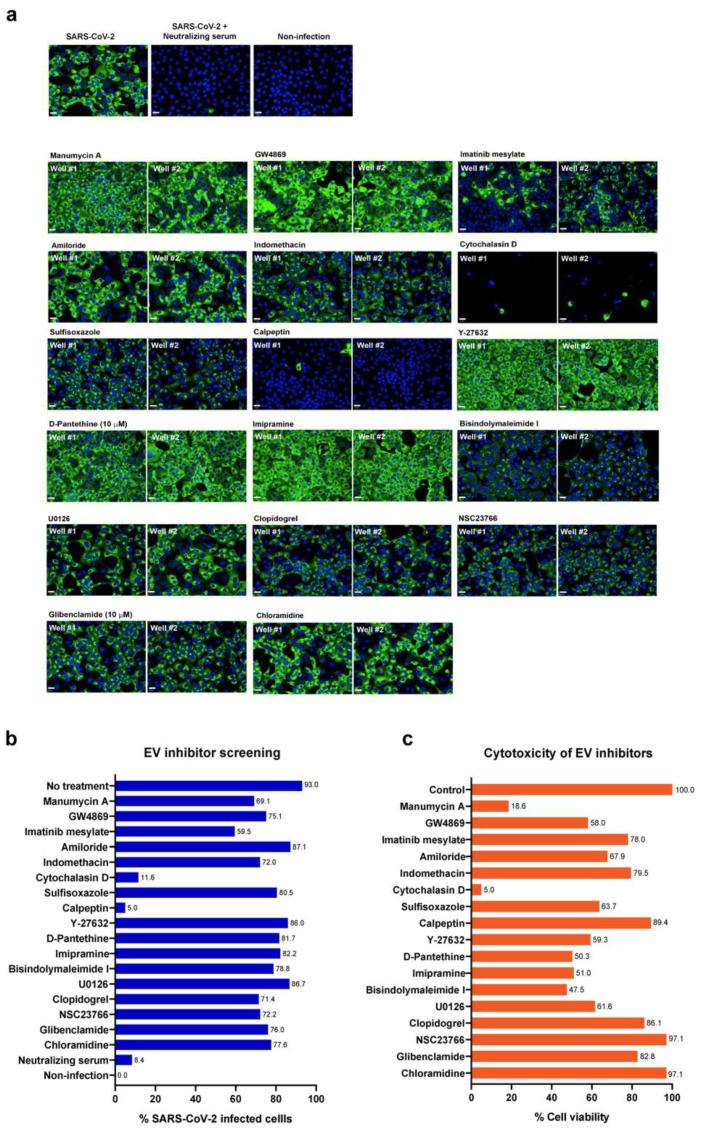
Screening of 17 known EV inhibitors against SARS-CoV-2 infected cells. Vero E6 cells were infected with SARS-CoV-2 at 25TCID50 for 2 h and subsequently treated with EV inhibitors at post-infection phases for 48 h. Positive convalescent serum of a COVID-19 patient was included as a positive control, and mock infection was performed in parallel as a negative control. The infected cells were then fixed and stained for viral nucleoproteins with anti-SARS-CoV NP mAb. The SARS-CoV-2 infected cells were detected by high-content imaging. (**a**) The high-content images of the infected Vero E6 cells treated with indicated EV inhibitors at 10 µM are shown. Fluorescent signals: green, anti-SARS-CoV NP mAb; blue, Hoechst. (**b**) The percentage of the infected Vero E6 was calculated for each condition. The data are presented as an average of two independent experiments. (**c**) Vero E6 cells were seeded in a 96-well plate overnight and then treated with 10 µM of indicated compounds in a medium for 48 h. Cell viability was examined by the MTT assay. Absorbance was measured at a wavelength of 570 nm. Data were normalized to the solvent control and presented as the percentage of cell viability. Scale bar: 20 µm.

**Figure 3 biomedicines-09-01230-f003:**
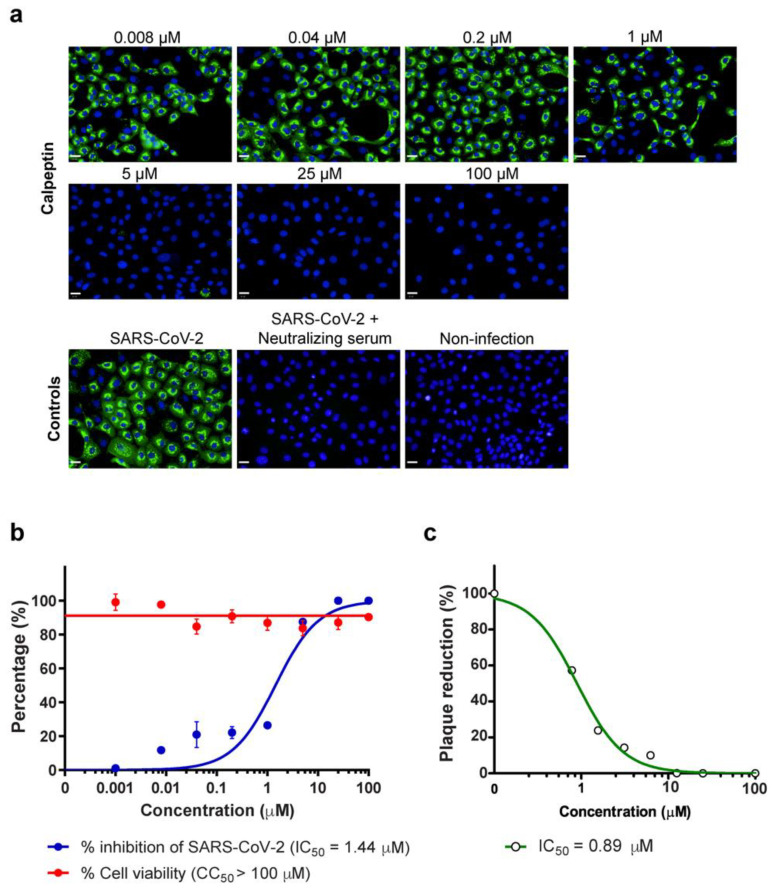
The dose-dependent effect, the half maximal inhibitory concentration (IC50), and the half maximal cytotoxicity (CC50) of calpeptin against SARS-CoV-2 infected Vero E6 cells. Vero E6 cells were infected with SARS-CoV-2 at 25TCID50 for 2 h and then post-infection treated with calpeptin, at concentrations ranging from 0.008 to 100 µM, for 48 h. Positive convalescent serum of a COVID-19 patient was included as a positive control. The supernatant was collected for viral output quantification. The infected cells were fixed and stained for viral nucleoproteins with anti-SARS-CoV NP mAb. The SARS-CoV-2 infected cells were detected by high-content imaging. (**a**) The high-content images of calpeptin treatment in SARS-CoV-2 infected Vero E6 cells are demonstrated. Scale bar: 20 µm. The percentage of inhibition was calculated as the percentage of the control conditions. The cytotoxicity assay was performed in parallel to evaluate the cell viability at each concentration. (**b**) The percentage of virus inhibition (blue) and cell viability (red) is shown. The data are presented as mean ± SEM of three biological replicates. (**c**) Viral output was examined by plaque reduction assay, and data are presented as % of the control (*n* = 2 biological replicates).

**Figure 4 biomedicines-09-01230-f004:**
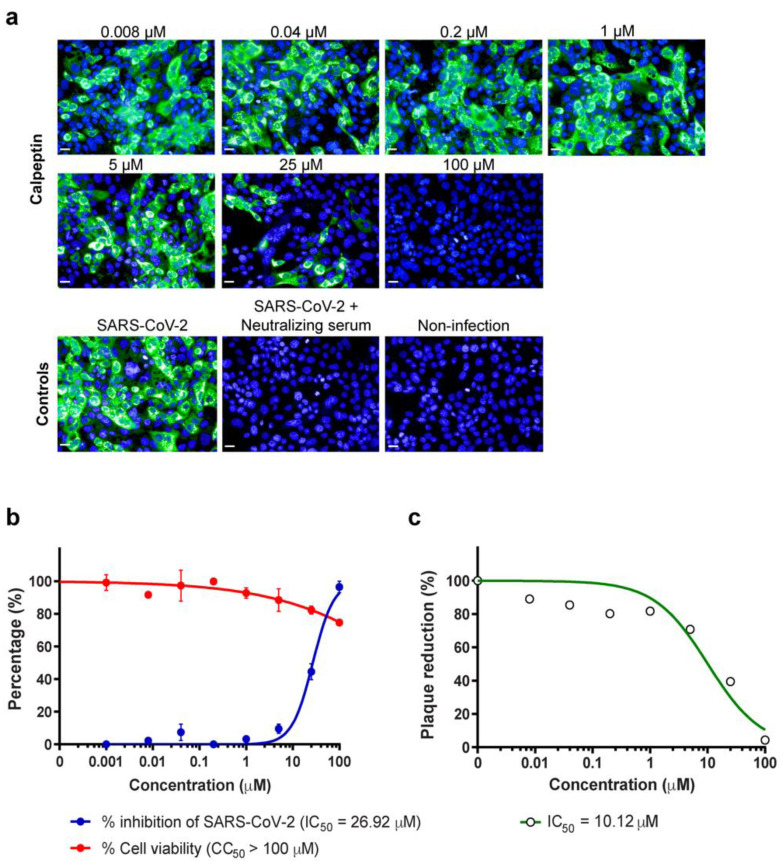
Anti-SARS-CoV-2 activity of calpeptin in human-lung epithelial cells Calu-3. Calu-3 cells were infected with SARS-CoV-2 at 25TCID50 for 2 h and then post-infection treated with calpeptin at concentrations ranging from 0.008 to 100 µM for 48 h. Positive convalescent serum of a COVID-19 patient was included as a positive control. The supernatant was collected for viral output quantification. The infected cells were fixed and stained for viral nucleoproteins with anti-SARS-CoV NP mAb. The SARS-CoV-2 infected cells were detected by high-content imaging. (**a**) The high-content images of calpeptin treatment in SARS-CoV-2 infected Calu-3 cells are demonstrated. Scale bar: 20 µm. The percentage of inhibition was calculated as the percentage of the control conditions. The cytotoxicity assay was performed in parallel to evaluate the cell viability at each concentration. The data are presented as the mean ± SEM of three biological replicates. (**b**) The percentage of virus inhibition (blue) and cell viability (red) is shown. (**c**) Viral output was examined by plaque reduction assay, and data are presented as % of the control (*n* = 2 biological replicates).

**Figure 5 biomedicines-09-01230-f005:**
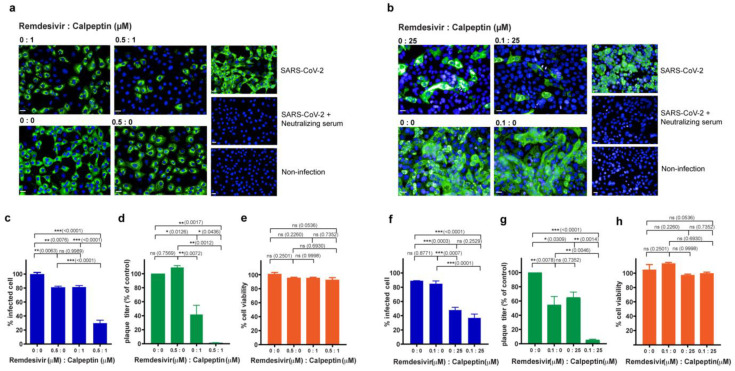
Effects of calpeptin and remdesivir combination on anti-SARS-CoV-2 activity in Vero E6 and Calu-3 cells. Cells were infected with SARS-CoV-2 at 25TCID50 for 2 h and then post-infection treated-indicated concentration of remdesivir (µM):calpeptin (µM). Positive convalescent serum of a COVID-19 patient was included as the positive control, and mock infection was performed in parallel as a negative control. The supernatant was collected for viral output quantification. The infected cells were fixed and stained for viral nucleoproteins with anti-SARS-CoV NP mAb. The SARS-CoV-2-infected cells were detected by high-content imaging. The high-content images of combination treatment in SARS-CoV-2 infected Vero E6 (**a**) and Calu-3 (**b**) cells are shown. The percentage of infected Vero E6 (**c**) and Calu-3 (**f**) was calculated and present at the indicated concentrations. The amounts of infectious virions in the supernatant of infected Vero E6 (**d**) and Calu-3 (**g**) cells were quantified by plaque assay, and data were presented as the percentage of the control. The percentage of cell viability of Vero E6 (**e**) and Calu-3 (**h**) are shown. The data are presented as the mean ± SEM of three biological replicates. Statistical analysis was performed by using one-way ANOVA with Tukey post hoc test: * *p* < 0.05, ** *p* < 0.005, and *** *p* < 0.001, ns, not significant. Scale bar: 20 µm.

**Figure 6 biomedicines-09-01230-f006:**
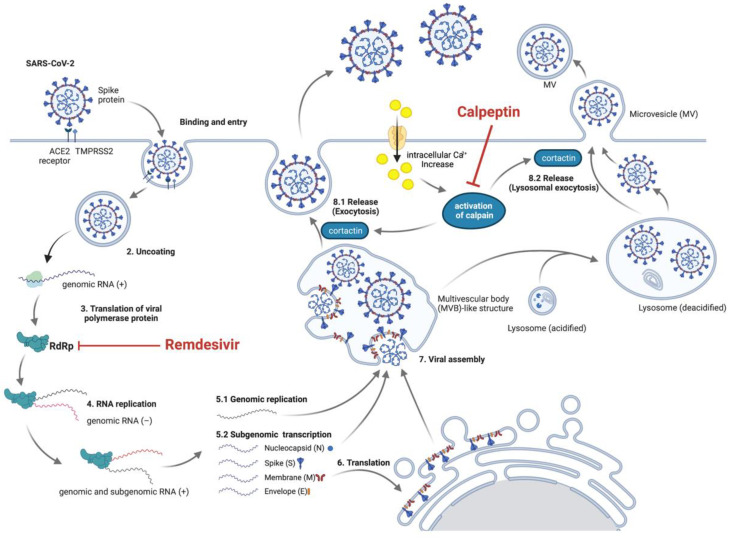
Proposed mechanisms of SARS-CoV-2 inhibition by remdesivir and calpeptin combination. Remdesivir, a virus-targeting drug, inhibits RNA-dependent RNA polymerase (RdRp), leading to the inhibition of SARS-CoV-2 replication. Calpeptin, a host-targeting compound, suppresses the viral release via inhibiting EV trafficking and shedding microvesicles, resulting in the inhibition of SARS-CoV-2 production and release. This figure was created with BioRender.com (accessed on 13 August 2021).

## Data Availability

All data are available in the main text or the Appendix A.

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
