# Peer review of "Anti-SARS-CoV-2 Activity of Extracellular Vesicle Inhibitors: Screening, Validation, and Combination with Remdesivir"

_biomedicines, 2021, doi:10.3390/biomedicines9091230_

Round 1
Reviewer 1 Report
In this study the authors report the identification of calpeptin as a potential treatment option for SARS-CoV-2. The study is of high interest and the experiments have been conducted thoroughly.
Some open questions remain:
It would be important to have information on the purity of the tested substances. Please provide this, e.g.analysis certificates of the vendors.
The authors write that the “human respiratory epithelial lining is the main site of SARS-CoV-2 infection”. Why they then used Vero E6 cells for the primary screen and not directly the Calu-3 cells? A thorough discussion on the significant differences between Vero E6 and Calu-3 cells is missing. Also the relatively low effect in Calu-3 cells (IC50 > 25 uM) should be discussed and compared to efficacies of other known drugs targeting SARS-CoV-2, for example compounds reported in reference 33. Also the IC50 of the calpeptin activity against Mpro should be compared with the data of this study (ref 34).
Please provide information on the vendor of DMSO at the first time, it is mentioned (l. 126)
What is 25TCID50 (l. 136?)
Figure 1: It would be better to write the effect of the tested compounds in a separate table instead into the figure caption.
Figure 2: There is no need to mentioned the dose at each subfigure. Also one digit after the comma is sufficient in 2B & C.
Figure 5: In 5a the yellow text is very difficult to read, please choose another color or place the numbers outside of the microscopic image. The label should be better Remdesivir : Calpeptin (uMol).
The text in c-f is difficult to read, please increase the font size.
l. 262: This should be moved to discussion section:
These results indicated that 262 calpeptin exerted antiviral activity against SARS-CoV-2 in a cell-type independent manner, suggesting that it might be a candidate for developing an effective and safe antiviral agent.
l. 297: This should be moved to discussion section:
This finding supports future investigations of calpeptin and remdesivir combination in pre-clinical animal models and early phase clinical studies.
The text contains many smaller errors, some are listed below. The authors should carefully go though the text:
l. 55 based on the in vitro and in vivo against SARS-CoV-2
l. 70: ...ligands and receptors, and cytosolic proteins (maybe: ...as well as cytosolic proteins?)
l. 77: utilized tetraspanin-enriched microdomain
l. 80: It has prospected that EV biogenesis and release modula- 80 tions can benefit RNA viral infection, including SARS-CoV-2. (better: It has been proposed), reference required
l. 84: As virus and EVs (better: EV)
l. 87: EVs inhibitors (better: EV inhibitors)
l. 93: and remdesivir was successfully performed to demonstrate (better: and remdesivir demonstrated…)
l. 108: SARS-CoV-2 virus (remove virus)
l. 117: will be performed (better: were cconducted
l. 150: followed by incubated with
l. 165: Sig-ma-Aldrich, St. Luis
l. 190: This study then focused (better: Therefore, this study focused…)
l. 191: could exert the anti-SARS-CoV-2 (better: exert an)
l. 192: were summarized (better: are shown)
l. 203: to primary screening the EV inhibitors (better: for a primary screen of the 17 EV inhibitors…)
l. 210: anti-SARS-Co-V2
l. 262: no apparent cytotoxic in Calu-3 cells
l. 322: Our finding was in line (better: Our findings are in line)
l. 357: This figure created by BioRender.com
Reviewer 2 Report
Here Sapusek Kongsomros et al report significant decline of SARS-CoV-2 infected cells at 48hpi of 1 (calpeptin) out of 17 EV-inhibitors added 2hpi. Importantly, they show additional effect when Calpeptide was used together with remdesivir. However, the dotplots seem to show that some (fig 3b and 4b) or all (fig 3c and 4c) are derived from one single experiment. Could the authors please only include data from at least three independent experiments.
Moreover, the cytotoxicity of the combined treatment of calpeptide+remdesivir is not shown and might void this promising treatment.
In the introduction a lot of space is used to explain the rational of using EV-inhibitors, and although this reviewer shares this sentiment the only EV inhibitor that showed significant effect is Calpeptin - a calpain cysteine protease inhibitor. Cysteine protease inhibitors have shown to block SARS-CoV-2 infection (A Clinical-Stage Cysteine Protease Inhibitor blocks SARS-CoV-2 Infection of Human and Monkey Cells | ACS Chemical Biology) but more specifically seem to block viral entry (Characterization of spike glycoprotein of SARS-CoV-2 on virus entry and its immune cross-reactivity with SARS-CoV | Nature Communications) by putatively binding RBD as indicated by preliminary results (Identification of Potent Small Molecule Inhibitors of SARS-CoV-2 Entry (biorxiv.org)) Therefore, to not mislead the reader please indicate that it might be due to entry inhibition rather than inhibition of EV formation-associated processes.
This reviewer applaud the authors for using a clinical isolate, but could the authors indicate what mutations are present in this isolate. The N501Y mutation has been shown to interfere with calpeptin binding to RBD.
Could the authors please add scale bars to the microscopy images.
In Line 91 'studing' should be 'studying'
Round 2
Reviewer 2 Report
The authors have greatly improved the manuscript. However at this stage there is insufficient experimental proof in this manuscript that demonstrates that inhibitors of EV machinery could be used as antivirals or as they claim that: "their study provided evidence to support the applications of EV inhibitors as antiviral agents". For example, most EV inhibitors in fig 2 show not much difference to their cytotoxicity, and to make stronger claims the authors should show the SD and provide statistics to show whether the other EV inhibitors besides calpeptin are statistically significant. Moreover, the authors will need to show accumulation of viral proteins or intact virions if they want to demonstrate that reduction of "viral output" (=titer after 48 hours) by calpeptin is specifically due to inhibition of virion production rather than effects on replication and entry effects.
